# Exploring the Significance of Pharmaceutical Care in Mental Health: A Spotlight on *Cannabis*

**DOI:** 10.3390/pharmacy12040100

**Published:** 2024-06-27

**Authors:** Claudete da Costa-Oliveira, Michele Lafayette Pereira, Nicole Ferrari de Carvalho, Luiza Aparecida Luna Silvério, Ygor Jessé Ramos, Priscila Gava Mazzola

**Affiliations:** 1Faculdade de Ciências Médicas, Universidade Estadual de Campinas, Campinas 13083-887, Brazil; 2APEPI—Medicinal Cannabis Research and Patient Support Association, Rio de Janeiro 20040-030, Brazil; 3Faculdade de Ciências Biológicas e Saúde, Universidade do Estado do Rio de Janeiro, Campus Zona Oeste, Rio de Janeiro 23070-200, Brazil; 4Faculdade de Ciências Farmacêuticas, Universidade Estadual de Campinas, Campinas 13083-871, Brazil; n165405@dac.unicamp.br (N.F.d.C.); luiza_luna_silverio@hotmail.com (L.A.L.S.); pmazzola@fcf.unicamp.br (P.G.M.); 5Farmácia da Terra Laboratory, Faculty of Pharmacy, Federal University of Bahia, Salvador 40170-115, Brazil; ygor.jesse@ufba.br

**Keywords:** *Cannabis*, anxiety disorders, depression, pharmaceutical care

## Abstract

Although preliminary evidence suggests *Cannabis*’s efficacy in symptom control for anxiety and depression—psychiatric disorders that significantly impact mental health—much remains to be understood about its effects on the central nervous system (CNS) and how to optimize treatment for these disorders. This study aims to conduct a narrative review to evaluate pharmaceutical care in treating symptoms of anxiety and depression alongside *Cannabis* use, focusing on safety and therapeutic efficacy optimization. We seek to conceptualize anxiety and depression disorders, review evidence on *Cannabis* use, evaluate the evidence quality, and identify knowledge gaps. Twelve articles were identified, revealing a significant gap in the literature regarding the integration of pharmaceutical care with *Cannabis*-based therapies, specifically for anxiety and depression. Despite a growing interest in the relationship between *Cannabis* and mental health, current research is insufficient for a comprehensive understanding. The relationship between *Cannabis* use and anxiety and depression disorders requires further, more targeted investigations. This study underscores the importance of future research to fill existing gaps, providing informed insights and robust guidelines for the safe and effective use of *Cannabis* as part of the treatment for anxiety and depression. It is crucial that pharmaceutical care integrates these therapies responsibly to improve the overall well-being of patients.

## 1. Introduction

In 2019, the World Health Organization (WHO) highlighted the widespread global prevalence of mental disorders, with an estimated 970 million people affected, including a significant proportion of adolescents [1]. Mental health issues such as anxiety and depression were identified as the most prevalent and are leading causes of disability worldwide. The economic impact is staggering, with mental health conditions costing the global economy approximately USD 2.5 trillion in 2010, a figure projected to escalate to USD 6 trillion by 2030 [2]. The societal impact is equally concerning, with neuropsychiatric disorders contributing significantly to work absenteeism and being linked to human rights issues, discrimination, and stigma [3].

Despite the significant burden, the pathophysiology of depression remains elusive, limiting the effectiveness of existing treatments. Current first-line treatments, including selective serotonin reuptake inhibitors (SSRIs) and cognitive behavioral therapy (CBT), do not suffice for all patients, with about a third not responding to pharmacotherapy [4,5,6]. This treatment gap persists even in high-income countries, underscoring the need for innovative therapeutic approaches [6,7].

*Cannabis sativa* L. and its derivatives have been increasingly researched for their therapeutic potential in mental health disorders, particularly focusing on the endocannabinoid system (ECS) and its integral role in modulating mood, cognition, and emotional responses [8,9,10,11,12]. Investigations into the ECS highlight the involvement of CB1 and CB2 cannabinoid receptors in potentially ameliorating symptoms associated with depression [6,13]. The ECS is pivotal in regulating various physiological functions, encompassing mood, cognitive abilities, and behavior [6,13,14]. It consists of the CB1 and CB2 receptors, endogenous lipid-based signaling molecules (endocannabinoids), and the enzymes that are responsible for their synthesis and breakdown. Key endocannabinoids, such as 2-arachidonoylglycerol (2-AG) and anandamide, are synthesized in a neuron-activity-dependent manner, traverse the synaptic gap in reverse (retrograde signaling), and activate CB1 receptors on presynaptic neurons to modulate neurotransmitter release. The widespread presence of CB1 receptors in the brain’s limbic system suggests a significant influence of endocannabinoids on emotional and cognitive functions. Furthermore, these receptors play a critical role in neuronal plasticity [14].

Recent studies have broadened the scope of investigation to include the multifactorial nature of depressive disorders and the ECS’s contribution to neuropsychiatric conditions [15]. Findings indicate that the ECS substantially influences neurotransmission, as well as the neuroendocrine and neuroimmune systems, all of which exhibit abnormalities in depressive states [15]. These insights underscore the potential significance of the ECS in the context of depression, positioning it as a potentially valuable target for novel therapeutic approaches [15]. Additionally, preliminary research on cannabidiol (CBD) suggests its promise as a safer option in psychiatric care, although further studies are essential to fully elucidate its therapeutic efficacy and safety profile.

This narrative review aims to critically examine the role of pharmaceutical care in the treatment of anxiety and depression with *Cannabis*. By emphasizing the pharmacist’s role in patient health, this study seeks to contribute to improving the quality of life of those who are affected by these conditions. Through a comprehensive literature search and analysis, this work addresses the current evidence on *Cannabis* use in treating anxiety and depression and identifies knowledge gaps, setting the stage for optimizing pharmaceutical care in this evolving field.

## 2. Materials and Methods

This review was conducted to answer the research question (RQ) of “How can pharmaceutical care be optimized in the treatment of anxiety and depression using *Cannabis*, aiming to increase the safety and efficacy of the treatment?”

The selected review methodology was narrative-based, predicated upon the observed prevalence and diversification attributed to variances in study timelines, reported outcomes, research focus, sample structures, data collection methodologies, and target demographics discerned within the scientific literature. The adoption of the Preferred Reporting Items for Systematic Reviews and Meta-Analyses (PRISMA) guidelines served to augment the transparency and precision in the result presentation (see Appendix A). The articles were systematically retrieved from specialized pharmaceutical journals, *American Journal of Pharmaceutical Education* (AJPE) and *Currents in Pharmacy Teaching and Learning* (CPT&L). They were searched in databases using the descriptors of “*Cannabis*”, “Anxiety”, “Depression”, and “pharmaceutical care”, associated with the boolean operators “AND” and “OR”.

Data collection took place from 7 July to 14 July 2023, and three research databases were used to collect the relevant studies: LILACS, PubMed, and SciELO. These databases were chosen because they are widely recognized in the health field and cover a wide range of scientific journals and academic literature.

### 2.1. Eligibility Criteria

This study adopted specific inclusion criteria to select relevant articles. Both in vitro and in vivo studies investigating the use of *Cannabis* as a treatment for anxiety and depression were considered. Additionally, studies involving patients with a confirmed diagnosis of anxiety and/or depression, and studies reporting outcomes related to the efficacy and safety of *Cannabis* use for treatment were included. Another inclusion criteria utilized was the involvement of pharmaceutical care in any of these described scenarios. Studies with different research designs, such as randomized clinical trials, cohort studies, and cross-sectional studies, among others, were admitted. This methodological variety allowed for a comprehensive analysis of the data, considering different perspectives and approaches in the investigation of the therapeutic use of *Cannabis* for anxiety and depression. For greater accessibility, only studies with free access to the full text were analyzed.

### 2.2. Exclusion Criteria

On the other hand, exclusion criteria were applied to select articles that presented inadequate research designs, were not available in peer-reviewed scientific journals, had small samples, or did not present relevant data for the review or did not meet any inclusion criteria. Relevant data were extracted from the searched articles, including information on research methods, the studied population, the intervention on *Cannabis*, the reported results, and the conclusions of the studies. The results of the studies were analyzed and synthesized to address the RQ of the review.

### 2.3. Article Processing Procedure

The review process entailed a meticulous examination of the full texts of the articles that met the initial selection criteria, undertaken by a dedicated review team. To ensure comprehensive coverage and minimize individual bias, each article was critically analyzed by a minimum of two, and typically three, members of the research team, who independently assessed the entire content of each paper. This approach was consistent across all articles to maintain uniformity in the review process.

In addressing the critical issue of publication bias—a common challenge in reviews—a thorough and systematic evaluation was performed to identify and mitigate potential biases in the selection of studies for inclusion. This step was crucial in recognizing any skewed representation of studies that could influence the review’s outcomes [16].

This methodological strategy was carefully designed to provide a well-rounded and equitable examination of the available evidence. By doing so, it aimed to significantly reduce the influence of publication bias, thereby ensuring that the conclusions drawn from this review are based on a balanced and accurately represented body of research [16].

After careful critical reading and a detailed team discussion, we organized the findings into distinct categories using a structured categorization methodology. This process identified and grouped key emerging themes, forming four discussion categories. It enabled a deeper, systematic analysis, offering a comprehensive view of the current knowledge in the field of study. This categorization enriched our understanding of the topic’s nuances and complexities, laying a foundation for discussing practical implications and guiding future research. Thus, this categorization method was crucial in cohesively synthesizing the review results, allowing us to present a detailed, informed analysis.

## 3. Results

The search results indicated that no articles corresponding to the searched keywords were found in the Scielo database. However, 10 articles were identified in the PubMed database, and 2 articles were found in the Lilacs database addressing the topic under consideration. Table 1 below displays the articles retrieved in the search.

Mental disorders, especially anxiety, present a complex clinical panorama in which the choice of appropriate treatment is influenced by multiple factors. Resistance to psychotherapy, the complex nature of the disorder, its severity, and whether it is concomitant with depression are determining factors in the therapeutic decision [29]. The main purpose of this manuscript was to map studies that associated innovations in *Cannabis*-based treatments with pharmacotherapeutic practices. The search resulted in 13 articles (see Table 2), a number that is reflective of the scarcity of research correlating pharmaceutical care with a therapy that has gained increasing prominence on the national and international scene [30].

However, further analysis revealed that the studies did not fully align with the pre-established inclusion criteria. The absence of research focused on pharmaceutical care practices targeting *Cannabis*-based treatments specifically for anxiety and depression symptoms was explicit. Although some studies [21,26] point to the relationship between *Cannabis* and mental health, the approach is secondary.

## 4. Discussion

Based on the inclusion and exclusion criteria, four primary categories were established for the discussion of the theme and the answering of the article’s RQ through bibliographic analysis of all available materials, beyond the selected databases: Category I—The Controversial Aspect of *Cannabis* in Mental Health; Category II—The Relationship between *Cannabis* and Pharmacotherapy for Anxiety and Depression; Category III—Integrative Pharmaceutical Care in *Cannabis* Use for Anxiety and Depression Symptoms; and Category; IV—Future Implications and Recommendations for Pharmaceutical Care with *Cannabis* in Mental Health.

### 4.1. Category I—The Controversial Aspect of Cannabis in Mental Health

The relationship between *Cannabis* and anxiety and depression is complex and still poorly understood. Some studies suggest that chronic *Cannabis* consumption may increase the risk of developing anxiety and depression disorders in some people, while others suggest that *Cannabis* may help alleviate the symptoms of these conditions [38], as seen in Table 2.

A review study published in 2022 in the medical journal *The Lancet Psychiatry* [39] concluded that *Cannabis* use is associated with an increased risk of developing anxiety and depression disorders in some users, especially adolescents and people with a genetic predisposition to these disorders. Other studies, however, show that *Cannabis* can help relieve symptoms of anxiety and depression in some people [34].

According to Di Forti et al. [31], with the large numbers of recreational users came also concerns about the risks of using *Cannabis*. According to the authors, this concern arose in Europe from large epidemiological studies that reported that *Cannabis* use increases the risk of schizophrenia-like psychosis. However, these studies did not collect detailed data on the patterns of use or potency of the *Cannabis* used, which could be an important factor in moderating the associated risk. In their work, the authors sought to demonstrate how daily use of high-potency *Cannabis* (high concentration of THC—delta-9-tetrahydrocannabinol) is associated with a particularly high risk of psychosis and other conditions. The most striking finding is that patients with a first episode of psychosis preferentially used high-potency *Cannabis* preparations, suggesting that the consumption of strains with high concentrations of THC and lower concentrations of CBD would be linked to an increased chance of outbreaks. However, the study stated that there was no significant difference between cases and controls regarding whether they had previously used *Cannabis* or the age of first use

In the study by D’Souza et al. [32], the authors also suggest an association between psychosis and the use of cannabinoid compounds. They also point out the scarcity of data directly evaluating the psychotomimetic effects of cannabinoid compounds and, in particular, THC. In their study, the authors sought to characterize the psychotomimetic effects of THC in healthy individuals selected under double-blind, placebo-controlled laboratory conditions, using standardized behavioral and cognitive assessments. The main finding of the study is that THC produced transient effects in healthy individuals, including suspicion, paranoid and grandiose delusions, conceptual disorganization, and illusions. It also caused depersonalization, derealization, distorted sensory perceptions, altered perception of the body, a sense of unreality, and extreme slowing down of time. THC produced negative symptoms, including blunted affect, reduced rapport, lack of spontaneity, psychomotor retardation, and emotional withdrawal. The authors report that various studies have shown that *Cannabis* use impairs various aspects of cognitive function in a dose-related manner, with deficits in verbal recall, and that their results are consistent with these effects. Comparing the data with other studies, the authors concluded that factors need to be considered, including the degree of use (tolerance) and lifetime exposure to *Cannabis*, as it is possible to find studies that found minimal effects of THC on cognitive test performance in subjects who were dependent on *Cannabis* and smoked an average of four *Cannabis* cigarettes a day for several years. The work did not provide conclusive answers.

Pharmacological inhibition and genetic manipulation targeting the CB1 cannabinoid receptor are known strategies that can mimic certain characteristics of melancholic depression. These methods specifically inhibit the receptor’s function, either through chemical agents that block receptor activity or by genetically modifying the expression of the receptor itself. This inhibition can lead to a state resembling melancholic depression, characterized by symptoms such as decreased appetite, heightened anxiety, increased arousal and wakefulness, difficulty in extinguishing aversive memories, and an increased sensitivity to stress [15,40,41].

According to Robinson et al. [40], the relationship between *Cannabis* uses and the incidence of psychotic outcomes as a causal factor is complex, influenced by the multifactorial nature of these outcomes, which are shaped by the environment, genes, and their interactions. The authors state that recent studies emphasize the importance of genetic influences, which may or may not increase the vulnerability of affected individuals to the psychogenic effects of *Cannabis* [40]. Although there is preliminary evidence suggesting that *Cannabis* can be effective in controlling symptoms of anxiety and depression and improving quality of life, more research is needed to determine whether *Cannabis* can be used as an effective treatment for symptoms of depression [6,7,37]. Freeman et al. [33] used a variety of methods, including real-life social situation tests, virtual reality immersion, and standard self-reporting and interview measures, to investigate whether an acute dose of intravenous THC causes paranoia in healthy adults with persecutory ideation [33]. The results showed that THC significantly elevated anomalous experiences, anxiety, worry, depression, and magical thinking. The study suggests that paranoia caused by THC may be mediated by cognitive mechanisms such as selective attention and a biased interpretation of social stimuli.

Experimental studies [42,43] suggest that CBD can partly neutralize the demotivating effects of THC in healthy individuals (men and women). The ratio of CBD to THC used was 1.25:1. However, to validate these results, it is necessary to investigate more classic depressive symptoms, such as depressed mood and anhedonia. Although there are many reports of humoral improvement with the use of CBD, there is still little evidence of the antidepressant effects of cannabinoids in humans. In addition, it is necessary to evaluate the effects of combining CBD and THC in animal models that are predictive of antidepressant activity, such as the forced swim test (FST) and the tail suspension test (TSC) in rats. The interaction of CBD with 5-HT1A receptors in the periaqueductal gray matter produced anxiolytic effects and facilitated the neurotransmission process. It has also been found that CBD has antidepressant effects at an effective dose of 30 mg·kg^−1^, provided there is activation of 5HT1A receptors [42,43]. Lafaye et al. [34] provide information on some important research on the subject; contrary to the effects of THC, some studies have shown that CBD can have anxiolytic effects [18]. In individuals with anxiety, CBD can considerably reduce symptoms compared to placebo. A clinical trial in healthy volunteers showed that the acute administration of CBD seemed to reduce experimentally induced anxiety without modifying baseline anxiety levels; it also seemed to reduce social phobias [44,45,46].

The treatment of symptoms from mental conditions such as anxiety and depression was a commonly reported reason for patients who do not have a doctor’s prescription for *Cannabis* use [47]. It has long been proposed that CBD inhibits THC-associated anxiety by antagonizing the cannabinoid receptor activation by THC [48]. CBD may also reduce anxiety via serotonin 5-HT1A receptors and/or GABAA receptors [35]. These receptor pathways are being explored in the hope of innovative therapeutic strategies for phobias, post-traumatic stress disorder, and drug abuse [49,50]. According to the authors, CBD not only showed a faster onset of action compared to classic antidepressants but also triggered an immediate antidepressant effect, which was similar to that reported for Ketamine for anxiety or depression [49,50]. In relation to depression, *Cannabis* can help relieve symptoms in some people, but the exact mechanism by which this occurs is still unclear. THC can increase the availability of neurotransmitters such as serotonin and dopamine, which are related to mood and well-being [38]. However, more research is needed to better understand this relationship. Bueno, Ortiz’s (2021) study supported the use of CBD in an acute and “chronic” (two-week) regimen, measuring its anxiolytic/antidepressant effects in behavioral and operational models [38]. The only observed side effects were a reduction in sucrose preference and reduced food consumption and body weight in non-operated animals treated with CBD.

Smith’s work from 2019, although not strictly in line with the selection criteria, raises pertinent reflections. Her study on the medicinal use of *Cannabis* among Canadian university students reiterates the complexity of the relationship between medical authorization, medicinal use, and recreational purposes. These data highlight the immediate need for rigorous pharmacotherapeutic monitoring, given the possible adverse consequences of *Cannabis*, such as intensifying symptoms of anxiety, causing panic attacks, and influencing the onset and evolution of psychoses, considering the dose and existing predispositions [18,31].

In conclusion, the relationship between *Cannabis* use and anxiety and depression disorders is an area that still needs further and more detailed investigation. The variability of results may be influenced by factors such as the potency of the *Cannabis*, the frequency and amount of use, individual sensitivity, and genetic predisposition. Furthermore, a deeper understanding of the diverse array of cannabinoids is warranted. Given this complexity, it is essential that further research, especially controlled clinical trials, is conducted to elucidate the specific effects of *Cannabis* on mental health. In the meantime, it is essential that individuals interested in using *Cannabis* to treat mental health problems seek appropriate guidance to assess the potential risks and benefits for each particular case.

### 4.2. Category II—The Relationship between Cannabis and Pharmacotherapy for Anxiety and Depression

Strickland et al. [21] focused on the relationship between the use of CBD-based products and the quality of life and mental health of patients with epilepsy. The research pointed out that, while there is an indication of a beneficial relationship between *Cannabis* and mental health symptoms, the current literature still has gaps that require more robust investigation [21,26].

Gershoni et al. [26] undertook a secondary analysis of data on patients who sought *Cannabis* for chronic pain relief. The study not only sought to assess perceived changes in pain intensity but also considered associated parameters such as sleep disturbance, quality of life, and, in a complementary way, symptoms of anxiety and depression. This suggests that any perceived improvement in mental parameters may be an indirect consequence of pain reduction and not necessarily a direct effect of *Cannabis* on the endocannabinoid system [26].

In Table 3, it is possible to observe works that explore the relationship between *Cannabis* and pharmacotherapy for symptoms of anxiety and depression, often providing information on drug interactions.

Like pharmaceutical drugs, cannabinoids such as THC and CBD have potential drug interactions due to CYP450 enzyme activity and protein binding. Few dedicated drug interaction studies have been carried out [53], but pharmacists can use their knowledge to recognize whether drugs are enzyme substrates, inducers, or inhibitors to prevent clinically significant interactions from occurring.

Side effects are mediated by the dose, route of administration, frequency of use, and other individual factors such as age and genetic predisposition, but the list is much less exhaustive than that observed in the pharmacotherapy of anxiety and depression disorders but needs to be widely known by pharmacists in order to provide accurate information. Prescribers often start *Cannabis* as a third- or fourth-line adjuvant therapy in addition to other drugs, increasing the potential for drug interactions and adverse effects [54].

In Table 4, common types of medications that are metabolized by enzymes that also interact with THC are highlighted, along with their potential clinical impacts, inhibitions, and interactions, revealing the synergistic or adverse effects of these interactions.

Some studies suggest that *Cannabis* can increase the sedative effects of drugs such as benzodiazepines, which are commonly prescribed to treat anxiety. This interaction can increase the risk of excessive sleepiness, confusion, and cognitive impairment [55].

**Table 4 pharmacy-12-00100-t004:** Enzyme substrates, inhibitors, and inducers. Adapted from “Potential adverse drug events with tetrahydrocannabinol (THC) due to drug–drug interactions.” Brown [56].

CYP3A4		CYP2C9
Substrates	Immunosuppressants, antidepressants, antipsychotics, opioids, benzodiazepines, statins, many others.	Antidepressants, antiepileptics, proton pump inhibitors, warfarin.
Impact	Increased substrate bioavailability, increased adverse effects of substrate.	Increased substrate bioavailability, increased adverse effects of substrate.
Inhibitors	Protease inhibitors, ketoconazole, nefazodone, amiodarone, verapamil, cimetidine, imatinib, tamoxifen.	Fluvoxamine, fluoxetine, proton pump inhibitors, ketoconazole, clopidogrel, fluconazole, fluorouracil (5-FU).
Impact	Increased THC bioavailability, increased THC adverse effects.	Increased THC bioavailability, increased THC adverse effects.
Inducers	Phenytoin, carbamazepine, topiramate, rifampicin, pioglitazone.	Rifampin, carbamazepine, phenobarbital, phenytoin, St. John’s Wort.
Impact	Decreased THC bioavailability, decreased effectiveness.	Decreased THC bioavailability, decreased effectiveness.

A recent meta-analysis emphasized that the use of cannabinoids is associated with a significant increase in the likelihood of neuropsychiatric side effects. While *Cannabis* shows short-term benefits in reducing neuropsychiatric conditions, prolonged use may exacerbate depression. Additionally, the effects of THC can be amplified or reduced by other common medications, increasing the toxicity by restricting metabolic pathways. These risks are more pronounced in medically complex patients, who are the focus of medicinal marijuana programs. *Cannabis* consumption necessitates careful management of patients with pre-existing cardiovascular diseases due to the potential for adverse effects such as hypotension, hypertension, syncope, and tachycardia. Additionally, THC may impair the body’s immune response, elevating the need for heightened monitoring, especially in patients undergoing immunosuppressive treatment [56].

An impact on cognition and motor skills can also be associated with *Cannabis* use. THC, in particular, can affect short-term memory, attention, and the ability to make decisions. These effects can be even more significant in adolescents, whose brains are still developing. Another common side effect of *Cannabis* is reduced motor coordination, which can increase the risk of car accidents and other injuries. In addition, chronic *Cannabis* use can also cause lung damage and other respiratory complications [10]. The work by Brown et al. [53] addresses an important aspect of the interaction between *Cannabis* and other drugs regarding its influence on Phase II metabolic pathways, more precisely on the enzymes of the UGT family (uridine 5′-diphosphoglucuronosyltransferase) [53]. These enzymes play a crucial role in the elimination properties of foreign substances from the body, including drugs. When CBD inhibits these enzymes, there can be a reduction in the elimination of drugs, leading to a build-up of these substances in the body. This accumulation can result in an increase in the plasma concentrations of other drugs, potentiating their pharmacological effects or increasing the risk of unwanted side effects. It is important to note that this interaction can vary between individuals, and some patients may be more susceptible to these metabolic changes than others [53].

In addition to specific medications for symptoms of anxiety and depression, the authors also point out that recreational users or consumers of CBD should also exercise caution when using the compound in conjunction with other common medications. Many healthy people use medicines such as contraceptives, painkillers, or antihistamines in everyday situations, which makes it important to consider the possibility of drug interactions with CBD. It is essential to bear in mind that recreational use of CBD may not be supervised by a health professional, which makes these interactions even more risky [53,54,55].

### 4.3. Category III—Integrative Pharmaceutical Care in Cannabis Use for Anxiety and Depression Symptoms

Pharmacists play a pivotal role in healthcare, particularly in contexts requiring careful management of treatments, such as *Cannabis* use for mental health conditions. Their expertise in medication management, patient education, and therapeutic optimization positions them uniquely to guide patients safely and effectively [57,58,59].

Reformulating the individual, social, and structural factors that influence mental health is a complex task that goes beyond the health sector. In this context, pharmaceutical care emerges as a fundamental approach to effective promotion and prevention in mental health, integrating efforts from multiple sectors. The pharmacist, as a health professional who is accessible and widely distributed in the community, plays a crucial role in integrating mental health promotion and prevention actions. They are able to provide precise and personalized guidance on the safe and appropriate use of psychotropic drugs, ensuring the effectiveness of treatment and reducing associated risks [57]. In Table 5, we can observe a significant decrease in pharmacist jobs related to the role linked to mental health and *Cannabis* issues.

At the same time, the administration of *Cannabis* as a medicine requires a careful and precise approach due to the various properties of the cannabinoids present in the plant, which can affect its pharmacological activity when ingested. According to Samanta et al. [58] the misconception that CBD does not cause adverse effects may be attributed to its natural origin. Medical supervision is necessary to monitor and prescribe the correct dosage, identify potential side effects, and assess potential drug interactions [58].

As experts in medicines, pharmacists are well equipped to provide clinical advice to patients and adequate supervision in the management and dispensing of safe medicines. They have the necessary knowledge to mitigate the potential risks associated with new therapies, including harmful drug interactions, contraindications, and potential addictive behavior, as well as having an existing infrastructure to handle controlled substances and a secure supply chain to limit diversion [59]. Pharmacists, with their expertise in pharmacology, are highly qualified to guide and monitor patients, ensuring that the dosage and combination of cannabinoids is appropriate for each individual, taking into account their specific therapeutic needs and reducing the risks of unwanted effects, as well as drug interactions, both when using OTC (over-the-counter) and polypharmacy.

The pharmacist’s view of healthcare needs to be broadened and go beyond the frontier of the health professional–patient relationship, meeting and considering the needs of the caregiver/companion. There is a dialogue with the user, but this often does not extend to the companion or caregiver. However, addressing the patient in their overall care environment is essential for the success of pharmacotherapy.

An assessment of previous medical history is essential before any decision is made to prescribe *Cannabis*, allowing for a comprehensive assessment of individual risks, comorbidities, and possible drug interactions. Cho et al.’s [23] research clearly illustrates this need by linking problematic health behaviors in adolescence to subsequent inappropriate use of opioids [23]. Therefore, the choice of treatments, whether traditional or innovative, can be improved to be more efficient and safer by strengthening the relationship between patients and health professionals. In this context, the patient’s history and particularities become crucial for informed decisions about specific treatments.

Healthcare professionals must have the ability to build a relationship of trust with the patient, to listen carefully to them, and to understand their needs. They must also be able to ask open and specific questions, synthesize information, and develop a treatment plan. Interviewing is a powerful tool that can help healthcare professionals provide quality care to their patients. As such, the ability to conduct interviews becomes invaluable in the life of a healthcare professional. Patients with syndromes such as depression are often associated with affective symptoms, changes in self-worth, ideational changes, and psychotic symptoms such as hypochondria, among others, conditions that require specific skills when carrying out an interview or consultation, which is both learned and intuitive and is a fundamental and irreplaceable attribute for health professionals when dealing with mental health, a fact that is often not taken into account or debated by pharmacists [62].

Through actions like these, pharmacists play an active and relevant role in comprehensive mental healthcare, integrating with different sectors to promote a unified and effective approach. With a focus on the patient, they can contribute significantly to improving the mental health of the community as a whole, guiding, supporting, and educating those who face challenges in this area, including those seeking treatment with *Cannabis* as a medicine [63]. Pharmaceutical care is the basis for ensuring the safe and effective administration of *Cannabis*, providing therapeutic benefits without the risks associated with self-medication and inappropriate use [51].

To ensure the safety of *Cannabis* use in the treatment of symptoms of anxiety and depression, monitoring side effects is essential. Strategies such as the use of standardized evaluation scales to quantify the intensity and frequency of symptoms can provide an objective assessment and guide treatment decisions [38].

The assessment of side effects should consider drug interactions, especially if the patient is taking other medications to treat symptoms of anxiety and depression [64]. Health professionals should be aware of possible side effects in different populations, such as the elderly, children, and pregnant women, making monitoring even more important. Brown et al. [56] highlight a crucial consideration regarding THC and other cannabinoids, emphasizing that the perception of these compounds, or a potential shift in it, is fundamental. Once recognized as medicinal substances, these compounds are regarded equally in terms of risks and benefits compared to prescribed products. This perspective can significantly influence the approach and acceptance of these substances in medical practice and society at large [56]. In this context, the role of pharmacists is fundamental in caring for patients seeking treatment with *Cannabis* for anxiety and depression symptoms. They provide guidance on the different types of *Cannabis* and their therapeutic effects, helping to choose the most suitable product [65]. In addition, pharmacists are responsible for calculating the correct dosages and monitoring patients to ensure treatment effectiveness and minimize side effects [51]. They also play an important role in identifying possible drug interactions, adjusting dosages, or suggesting alternatives when necessary [64]. In this way, the pharmacist’s collaboration in the management of *Cannabis* treatment for anxiety and depression symptoms is essential to provide safe and effective care for patients, contributing to the promotion of mental health in the community.

There is no doubt that the role of the well-known pharmacist and other professionals is fundamental, but it is important to stress that tools for the process of research, development, medicalization, and pharmaceutical care of *Cannabis* also need to exist and be available to support the scientific and medical community and the population in general [66,67]. One example is the document “Improving Medical Marijuana Management in Canada”, which talks about improving the management of medical marijuana in Canada [68]. It addresses issues related to the prescription, use, drug interactions, and other relevant clinical aspects of medical marijuana. The aim of the document is to provide information and guidance to pharmacy professionals so that they can offer high-quality and safe care to patients who use medical marijuana. The text discusses the clinical implications of using medical marijuana, including potential side effects, interactions with other medications, and safety measures to be considered. In addition, the document may also address issues related to public health policies, regulations, and strategies to ensure adequate access to medical marijuana for patients with legitimate medical needs.

In summary, the document seeks to provide a solid basis for the effective management of medical marijuana in the context of pharmaceutical practice in Canada, promoting safety, efficacy, and the best possible care for patients using this type of treatment. In the United States, although *Cannabis* is still a controlled substance, there are various bodies that provide information and guidelines on the plant, but the two main ones are the Food and Drug Administration (FDA) and the National Institute on Drug Abuse (NIDA) [69,70]. The FDA is responsible for regulating medical *Cannabis* products and other medicines in the United States. The FDA reviews scientific research and approves *Cannabis*-based medicines that meet certain safety and efficacy criteria.

NIDA is a component of the National Institutes of Health (NIH) and focuses primarily on research into drug abuse and its effects on health. Although it does not regularly make products or prescribe medications, NIDA conducts and funds scientific research related to medical *Cannabis* and its health effects. NIDA also provides information and resources for health professionals, such as pharmacists, and the general public on the use of *Cannabis* for medicinal purposes [70].

The National Institute for Health and Care Excellence (NICE) is an autonomous organization in the United Kingdom that provides guidance to the National Health System (NHS) and other health professionals. The organization reviews the available scientific evidence on the medicinal use of *Cannabis* and formulates guidance for healthcare professionals on how to approach treatments [71]. NICE guides future research in this area and conducts systematic reviews of existing research into the use of *Cannabis* for medicinal purposes, examining its efficacy, safety, and possible side effects. Based on these reviews, they develop practical guidance and recommendations for healthcare professionals, ensuring that they have access to the most up-to-date and reliable information on the use of medical *Cannabis* for certain medical conditions [71]. These guidelines help healthcare professionals make informed decisions about the risks and benefits and when and how to prescribe medicinal *Cannabis* products, ensuring that their use is safe and based on sound scientific evidence.

In summary, pharmaceutical care is crucial in promoting mental health and managing treatment with medical *Cannabis*. Pharmacists’ deep understanding of pharmacology qualifies them to determine the correct dosing and cannabinoid combinations, tailored to individual therapeutic needs. It is essential to monitor for adverse effects of *Cannabis*, including potentially dangerous drug interactions, especially in patients already receiving other medications for anxiety and depression. The ability of pharmacists to communicate clearly with patients and their caregivers is vital, ensuring the delivery of precise and personalized information about the safe and effective use of *Cannabis*. An in-depth evaluation of each patient’s medical history is critical before prescribing *Cannabis* to understand individual risks, comorbidities, and possible drug interactions. Pharmacists should collaborate with other health professionals to integrate medicinal *Cannabis* use into a unified and effective treatment plan. Staying updated with the latest research and guidelines on medicinal *Cannabis* is fundamental to ensure that care is based on robust evidence and safe practices. Additionally, pharmacists must carefully assess the benefits and risks of *Cannabis* use, adjusting treatments and suggesting alternatives when necessary to ensure the safety and efficacy of the treatment. Together, these efforts by pharmacists contribute significantly to improving community mental health through informed, safe, and effective pharmaceutical care.

### 4.4. Category IV—Future Implications and Recommendations for Pharmaceutical Care with Cannabis in Mental Health

The present study highlights the gap in the literature when it comes to pharmaceutical care for dispensing *Cannabis* to patients who need to address mental health issues. More than just pointing this out, it is necessary to make notes on what interventions could be important for this area of research. Table 6, adapted from Henman et al. [72], reveals the main issues highlighted in the literature, possible intervention methods, and what results could be expected from these actions. In different sectors, the pharmaceutical care professional faces challenges. Frank et al. (2018) emphasize that with the increase in medicinal *Cannabis* users, there is also a demand for pharmaceutical professionals who accompany or may accompany these patients to address issues such as effectiveness, safety, dosage, and drug interactions [60]. In their study, the authors, sought to identify pharmacy students’ knowledge about the use and dispensing of *Cannabis*-based medicine through interviews. Not surprisingly, there was a lack of confidence in the information they possessed, and 80% of them felt that the topic of medical *Cannabis* should be added to the academic curricula. In this scenario, the importance of developing guides and protocols to assist these professionals, not only in the context of *Cannabis* dispensing but also because of the importance of providing adequate care to individuals needing treatment for mental health issues [68], becomes evident. Vilallón et al. [73], in Argentina, developed a guide aiming to develop a guideline to provide practical recommendations for the dispensing of *Cannabis* in primary care [73]. In the study, the authors also make notes on how the lack of clinical evidence studies in this area hinders the work of health professionals. In the study, the aim of the text in RQ was to be simplified, easily accessible, and cohesive to be simplified, easily accessible, and cohesive. In the province of Jujuy in Argentina in 2021 [74], the executive branch developed a more comprehensive guide on the clinical management of *Cannabis*. Other important topics to assist in decision-making in clinical practice were pointed out, including a table and clinical context and the ideal composition of *Cannabis* for each of them and an ideal flowchart of the process of user registration and *Cannabis* dispensing in the region. Despite the different complexity of the guides, it is important to emphasize that both have indications about the importance of multidisciplinarity and integration among professionals in the decision-making and dispensing process of the medication. Interprofessional collaboration needs to be incorporated into practice in health services, and it is extremely important to help ensure coordination and continuity of service delivery. Continuous professional development for pharmacists supports growth and strengthening within their scope of practice, improves collaboration, and provides standardized, high-quality, and appropriately certified health services [72].

Multidisciplinarity is not only important in the dispensing of the medication; it is essential that the approach to patients facing challenging mental or physical conditions be assertive, sensitive, and efficient. As already addressed in this work, the authors Dalgalarrondo et al. [62], in the book *Psychopathology and semiology of mental disorders*, brought up important issues about the importance of a sensitive attitude when interviewing patients and families in primary care. However, it is necessary that these narratives be directed to all professionals in the care process, including the pharmacists. Samorinha et al. [75] conducted semi-structured interviews exploring the practices, challenges, and strategies to improve the care of community pharmacists in dispensing and counseling, the confidence and comfort of pharmacists in providing care, and attitudes and beliefs regarding mental illnesses, and the result pointed to many reports of discomfort on the part of professionals during encounters with patients with mental illnesses [75]. According to the authors, training in communication skills and patient-centered psychiatric therapy is necessary to improve the services provided by pharmacists, along with increased collaboration with other professionals [75].

In a descriptive cross-sectional survey of pharmacists in Nigeria [76], access to patients’ medical records (64.3%) was the main barrier to providing pharmaceutical care cited by them. In the USA, a study highlighted that patients significantly improved when pharmacists worked together with physicians and other healthcare professionals [77]. Pharmaceutical care also mitigated care fragmentation, reduced costs, and improved health outcomes. A lack of interaction among healthcare professionals and a shortage of prescribers or pharmacists can lead to ineffective care and underutilization of resources [77].

Polypharmacy, characterized by the use of multiple medications, which often occurs with elderly patients, is directly linked to an increased risk and severity of adverse drug reactions, vulnerability to cumulative toxicity, medication errors, reduced treatment adherence, and increased morbidity and mortality. Additionally, the prescription of inappropriate medications can become a risk factor [78]. In this scenario, regular medication reviews conducted by pharmacists and multidisciplinary teams can play an important role. These interventions not only optimize therapy by reducing polypharmacy and minimizing unwanted drug interactions but also significantly improve the patient’s quality of life [78].

In the elderly, reducing the impact of polypharmacy is a challenging and much-desired task. In Pessoa et al.’s [61] study, the authors pointed out that there is still a lack of research on the use of *Cannabis* in the elderly population and that it is still a topic to be debated, but they highlighted the importance of *Cannabis* as a therapeutic alternative for the treatment of the symptoms of dementia syndrome. The results of the study indicate high efficacy through the induction of neurogenesis and a reduction in TAU hyperphosphorylation, resulting in a decrease in polypharmacy in the use of opioids and anti-inflammatories, providing relief and reducing undesirable symptoms. The effectiveness of a combined treatment with few adverse effects considerably increased the quality of life of these elderly patients. The authors also highlight the importance of public attention to these issues; the subject still needs to be debated, analyzed, and researched so that new policies can be instituted to make their use more accessible to those who need it. According to the authors, criminalized use due to historical sociocultural issues such as racism interferes with making treatment accessible and common, as cultivation has low production costs, but bureaucracies hinder cultivation, often requiring importation and significantly limiting access.

In this bureaucratic sphere, demystifying the use of medical *Cannabis* is fundamental. The creation of public guidelines, aligned with current legislation and based on regulated research, can increase the number of professionals with expertise in this area, reducing the burden on professionals and ensuring broader and safer access for patients.

**Table 6 pharmacy-12-00100-t006:** Future implications and recommendations for *Cannabis* pharmaceutical care in mental health.

Sector	Problems	Interventions	Expected Impacts
Pharmaceutical Care	The lack of information about the posology and dispensing of medical *Cannabis* is a significant problem. The process of use, from selection, prescription, and dispensing to patient education, often occurs in a disjointed manner. This can lead to a lack of monitoring or inadequate monitoring, resulting in limited benefits and unintended consequences [29,60,68,73,74].	The creation of pharmaceutical guidelines and protocols, sharing the responsibility of patient care with the prescriber for these medications, along with the introduction of the subject in specialized curriculum.	The creation of a coordinated and structured process where, through increased patient-centered care provided by pharmacists and enhanced collaboration with prescribers, both teams and patients achieve the maximum expected benefit.
	A need for sensitive approach in interviews with anxious and/or depressed patients [62,75].	Training and protocols to ensure the information.	Increased support and establishment of trust among staff, patients, and family members.
	Fragmentation in primary care, due to a lack of interaction among healthcare professionals, shortage of prescribers, or pharmacy professionals, can result in ineffective care and/or underutilization of resources [23,76,77,79,80].	Pharmacists working in a multidisciplinary manner in their community practices with *Cannabis* patients can assist in dosage adjustment, treatment assessment, monitoring, history, and potential risk habits.	Improvement in effectiveness and proper resource utilization: enhancement in care coordination, treatment optimization, and reduction in risky behaviors.
	Elderly patients often receive multiple medications (polypharmacy) but may be evaluated infrequently for various reasons. Unnecessary, inappropriate, or outdated medications or unwanted drug interactions can reduce the patient’s quality of life and increase the likelihood of hospitalization [50,80].	Medication review by pharmacists and multidisciplinary team reviews to optimize or replace medications or therapies. Guidance and counseling for patients and care teams on medication use, monitoring, discontinuation, or substitution.	Reduction in polypharmacy and inappropriate medications or those with adverse effects, thereby reducing prescription cascades.
Public Health	Demystification of the use of medical *Cannabis* as a prohibited and difficult-to-access substance [61,69,70,71].	Creation of public guidelines for regulated research and information.	Increases the number of professionals with necessary expertise, reducing the number of overwhelmed professionals and patients without access.

This study, by integrating results with existing research on the use of *Cannabis* in treating anxiety and depression symptoms, underscores the methodological diversity and variability in study populations, dosages, and measured outcomes, leading to significant heterogeneity and limiting the generalization of results. Moreover, focusing on short follow-up periods prevents the assessment of long-term effects, including potential adverse effects and the sustained efficacy of treatment. The presence of potential biases in the analyzed studies, such as selection and publication bias, and attempts to mitigate the latter by including unpublished studies, still leaves gaps in capturing all relevant research. These limitations underscore the need for more rigorous research, such as randomized clinical trials with robust designs and longer follow-up periods, to provide more conclusive evidence on the effects of *Cannabis*. Consequently, while the findings offer valuable insights, they must be viewed cautiously due to the cited limitations. Yet, the lack of clinical evidence in published studies hinders health professionals’ work, emphasizing the need for public guidelines that are aligned with legislation and based on regulated research to demystify medical *Cannabis* use, reduce the professional burden, and ensure broader, safer access for patients. This integrated discussion clarifies the study’s limitations and the broader implications for healthcare practice and policy.

## 5. Conclusions

Since its creation, the term pharmaceutical care has guided the work of thousands of pharmacists around the world. It is considered to be the professional commitment to health education and the promotion of the rational use of medicines, with the aim of maximizing the patient’s therapeutic benefit from medication. This sheds light on the patient, rather than focusing solely on the drug [81].

This definition highlights the importance of a holistic approach in clinical practice, which not only emphasizes the promotion, protection, and recovery of health but also the prevention of harm. In the context of treating mental disorders, this implies a comprehensive understanding of the patient’s history, their individual needs, and their general mental health.

The research revealed a significant gap in the existing literature. Despite the growing interest in the relationship between *Cannabis* and mental health, none of the studies directly and comprehensively addressed the integration of pharmaceutical care with *Cannabis*-based therapies. This lack of detailed information highlights the urgent need for further research to fill this gap, providing healthcare professionals and patients with informed insights and robust guidelines for the safe and effective use of *Cannabis* as part of treatment for anxiety and depression symptoms.

Well-defined guidelines and protocols can be valuable tools for healthcare professionals, offering evidence-based guidance on the appropriate combination of *Cannabis*-based treatments and pharmaceutical care practices. These guides can cover not only the dosage and administration of *Cannabis*, but also the careful selection of patients, the ongoing monitoring of their mental health, and the assessment of potential risks and drug interactions.

In summary, as we explore the therapeutic possibilities of *Cannabis* for mental disorders, it is imperative to proceed with caution and base our decisions on solid data. The lack of studies integrating pharmaceutical care with *Cannabis* therapy highlights the importance of future research in this field. In this context, pharmaceutical care emerges as a fundamental principle for promoting mental health, ensuring that therapies are administered responsibly and contribute to patients’ overall well-being.

## Figures and Tables

**Table 1 pharmacy-12-00100-t001:** Data sources.

Article	Author	Year	Local	Eligible	Exclusion Criteria
Using New and Innovative Technologies to Assess Clinical Stage in Early Intervention Youth Mental Health Services: Evaluation Study	Ospina-Pinillos et al. [17]	2018	Australia	no	The study aimed to examine the efficacy of a novel intervention known as the Mental Health eClinic (MHeC) in identifying early clinical signs of mental health disorders among young individuals. However, it did not address the potential therapeutic use of *Cannabis* in youth mental health or explore the role of pharmacists in facilitating mental health services.
*Cannabis* Use for Medicinal Purposes among Canadian University Students.	Smith et al. [18]	2019	Canada	no	The study examines the medicinal utilization of *Cannabis* among university students, encompassing aspects such as prevalence rates, motives for usage, and substitution of conventional therapies. Nevertheless, a comprehensive analysis of the effectiveness and safety of *Cannabis* as a treatment option for anxiety and depression is lacking.
Exploring associations between early substance use and longitudinal socio-occupational functioning in young people engaged in a mental health service.	Crouse et al. [19]	2019	Australia	no	The study investigates the correlation between early substance use and socio-occupational functioning in youth seeking early intervention for mental health services. However, the article does not specifically explore the therapeutic use of *Cannabis* for anxiety and depression, and it also falls short in meeting other predefined criteria.
Prescription Stimulant Misuse and Risk Correlates among Racially-Diverse Urban Adolescents.	Goodhine et al. [20]	2020	United States of America	no	The study addresses the misuse of prescription stimulants among urban adolescents but does not focus on the use of *Cannabis* as a treatment for anxiety and depression. Additionally, it does not mention the involvement of pharmaceutical care in any of the scenarios described in the inclusion criteria.
Cross-sectional and longitudinal evaluation of the use of cannabidiol (CBD) product in people with epilepsy.	Strickland et al. [21]	2021	United States of America	no	The study focuses on evaluating the use of artisanal CBD products in people with epilepsy but does not investigate the use of *Cannabis* as a treatment for anxiety and depression.
Randomised, pragmatic, waitlist controlled trial of *Cannabis* added to prescription opioid support on opioid dose reduction and pain in adults with chronic non-cancer pain: study protocol.	Jashinski et al. [22]	2021	United States of America	no	The study does not address pharmaceutical care involvement and does not meet the inclusion criteria, as it does not focus on individuals diagnosed with anxiety and depression. Consequently, it does not align with the established requirements for selecting pertinent articles for the proposed analysis.
Behavioral Health Risk Factors for Nonmedical Prescription Opioid Use in Adolescence.	Cho et al. [23]	2021	United States of America	no	The study centers on high school adolescents from Los Angeles and examines the risk factors linked to the non-medical use of prescription opioids. Nevertheless, it does not specifically target individuals diagnosed with anxiety and depression, nor does it delve into the involvement of pharmaceutical care.
An initial analysis of the UK Medical *Cannabis* Registry: Outcomes analysis of first 129 patients.	Erridge et al. [24]	2021	United Kingdom	no	This study analyzes the initial outcomes of patients in the UK prescribed with *Cannabis*-based medicinal products, focusing on the effects on health-related quality of life and clinical safety. However, it does not specifically address individuals diagnosed with anxiety and depression or discuss the involvement of pharmaceutical care.
Consumer Experiences with Delta-8-THC: Medical Use, Pharmaceutical Substitution, and Comparisons with Delta-9-THC.	Kruger et al. [25]	2022	United States of America	no	The study addresses the use and consumption patterns of delta-8-THC, a compound found in *Cannabis* products, in the United States. However, it does not discuss the impact of using this compound on patients diagnosed with anxiety and depression.
Wellness of patients with chronic pain is not only about pain intensity	Gershoni et al. [26]	2022	Israel	no	The study examines the effects of medicinal *Cannabis* use in managing chronic pain, only meeting one of the inclusion criteria, which is the availability of the full text. To be included, the study must meet all criteria.
Practical pathway for the management of depression in the workplace: a Canadian perspective	Chokka et al. [27]	2022	Canada	no	The study addresses major depression in the workplace, recognizing it as a significant factor affecting employee productivity and well-being. Although it addresses depression and anxiety, it does not establish any correlation with *Cannabis*.
The Association of Vaping With Social/Emotional Health and Attitudes Toward COVID-19 Mitigation Measures in Adolescent and Young Adult Cohorts During the COVID-19 Pandemic.	Oliver et al. [28]	2023	United States of America	no	The study investigates the associations between vaping among adolescents and young adults and their social and emotional health during the COVID-19 pandemic. However, it does not explore the use of *Cannabis* as an alternative to vaping or any mental health conditions.

**Table 2 pharmacy-12-00100-t002:** Summary of included articles on the controversial side of Cannabis in mental health.

Author	Year	Location	Pathology	Study
Di Forti et al. [31]	2009	England	psychosis	case–control
D’Souza et al. [32]	2009	United States of America	psychosis	review
Freeman et al. [33]	2015	United Kingdom	paranoia	randomized clinical trial
Lafaye et al. [34]	2017	France	-	review
Lowe et al. [13]	2018	Canada	schizophrenia, mood disorders, and anxiety	review
Dienst et al. [35]	2019	United States of America	social anxiety	observational study
Smit et al. [18]	2019	Canada	-	cross-sectional study.
Guid et al. [36]	2020	Argentina	refractory epilepsy	opinion
Luan et al. [37]	2020	China	depression	review
Cho et al. [23]	2021	United States of America	-	prospective cohort study
Bueno et al. [38]	2021	Brazil	anxiety	review
Petrilli et al. [39]	2022	United Kingdom	depression, anxiety, psychosis, or *Cannabis* use disorder	review
Robinson et al. [40]	2023	Canada	psychosis	review

**Table 3 pharmacy-12-00100-t003:** Summary of included article on the relationship between *Cannabis* and pharmacotherapy for anxiety and depression.

Author	Year	Location	Pathology	Study
Hill et al. [41]	2005	Canada	Depression	review
Crippa et al. [46]	2010	Brazil	Anxiety	clinical trial
Bergamaschi et al. [45]	2011	Brazil	Anxiety	randomized clinical trial
Linge et al. [49]	2016	Spain	Depression	experimental animals
Corroon et al. [50]	2017	United States of America	Pain, Anxiety, and Depression	review
Silote et al. [43]	2019	Brazil	Depression	review
Dienst et al. [35]	2019	United States of America	Anxiety	observational study
Berger et al. [44]	2020	Australia	Anxiety	case
De Aguiar et al. [12]	2021	Brazil	Anxiety	review
Zanellati et al. [51]	2021	Brazil	Anxiety	review
Dubey et al. [9]	2021	Indian	Anxiety	experimental animals
Bueno et al. [38]	2021	Brazil	Anxiety	review
Gallego-Landin et al. [15]	2021	Brazil	Depression	review
Frias et al. [52]	2022	Brazil	Anxiety and Depression	review
Petrilli et al. [39]	2022	United Kingdom	Anxiety and Depression	review
Bright et al. [6]	2022	Israel	Depression	review
Sharafi et al. [7]	2022	Iran	Depression	review
Nguyen et al. [11]	2023	United States of America	Anxiety and Depression	data analysis

**Table 5 pharmacy-12-00100-t005:** Summary of included articles on the pharmacist’s role in *Cannabis* clinical care for anxiety and depression.

Author	Year	Local	Pathology	Study
Caligiuri et al. [60]	2018	United States of America	-	interview
Dattani et al. [54]	2019	Canada	-	review
Guido et al. [36]	2020	Argentina	refractory epilepsy	opinion
Pessoa et al. [61]	2021	Brazil	dementia syndromes	review

## Data Availability

Data is contained within the article or Appendix A.

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
