# Peer review of "Exploring the Significance of Pharmaceutical Care in Mental Health: A Spotlight on Cannabis"

_pharmacy, 2024, doi:10.3390/pharmacy12040100_

Round 1

Reviewer 1 Report

Comments and Suggestions for Authors

The review assesses current understanding of the associations between cannabis and mental health disorders. It looks at both cannabis' propensity to be involved in the development of anxiety and depression and also their treatment by cannabis. Additionally, the review discusses the present and potential future role of the pharmacist in cannabis interventions. Few studies were included due to the fact that not many reliable studies have been published. The authors include a lot of their own opinions about the patient-pharmacist relationship towards the end of the manuscript.

General concept comments: I think the article is generally fine as it is. I think it could benefit from the title and/or abstract being modified to make it clearer that the patient-pharmacist relationship is considered in detail.

Specific comments:

Line 25 - "A literature review was conducted to select relevant studies." Sentence could be deleted.

Line 130 - Missing a period "." at the end of the sentence.

Line 134 - Missing a period "." at the end of the sentence.

Line 139 - "field of study" arguably sounds better than "study field".

Table 1 - "Eligible" in English not "elegible"; I'm not sure what the purpose or the benefit is for the column titled "Exclusion Criteria" in which all the boxes below state the same text "Does not meet all inclusion criteria". In my opinion that column can be deleted as I don't think it adds anything useful. Alternatively more information could be added i.e. why don't those articles meet the inclusion criteria? Table 1 is not referred to anywhere in the main text. Finally, that table could be moved to the supplementary materials.

Line 152 - Recommend saying "and whether it is concomitant..."

Line 155 - Suggest rephrasing to "The search resulted in 12 articles, a  number reflective of the the scarce literature correlating pharmaceutical care with a therapy that has gained increasing prominence on the national and international scene [56]."

Line 195 - "In her work, the author sought to demonstrate how daily use of high-potency Cannabis (high concentration of THC - delta-9-tetrahydro- cannabinol) is associated with a particularly high risk of psychosis and other conditions. The most striking finding is that patients with a first episode of psychosis preferentially used high-potency Cannabis preparations, suggesting that the consumption of strains with high concentrations of THC and lower concentrations of CBD would be linked to an in-creased chance of outbreaks." Were these findings found to be statistically significant? If yes, comment. If no, comment.

Line 222 - "The possible pharmacological and genetic blockade of the CB1 cannabinoid receptor" Why possible? Is it only hypothesised or is it known? How do cannabinoids act as a genetic blockade? I'd like to see more elaboration on the information in this sentence.

Line 223 - "which would include symptoms such as reduced food intake" It is well-known that cannabis can induce hunger (the munchies). Can you elaborate on this? Has it been found that cannabis use decrease food consumption? Or is this just a guess?

Line 230 - delete the word "significant".

Line 280 - It should say "These data highlight" as data is plural and datum is singular. Note: In English people often use the word data as if it were singular e.g. "the data is", however, this is grammatically incorrect as the word derives from Latin.

Line 289 - It's perhaps worth mentioning that the compounds are not just THC and CBD but the array of compounds in cannabis which contribute to the "entourage effect".

Line 347 - "Cannabis consumption poses significant risks for individuals with pre-existing cardiovascular diseases, potentially leading to hypotension, hypertension, syncope, and tachycardia. Moreover, THC may compromise the body's immune response, increasing susceptibility to infections, particularly concerning for patients undergoing immunosuppressive treatment [15]." I'm not sure that I really believe that cannabis poses a "significant risk" regarding these conditions. I'd like to see a more critical evaluation of the source of all this information. If the evidence is strong, I'd like to know, if the evidence is weak then I'd like that to be dicsussed.

Category 3 - I like this section as it highlights the potential usefulness of pharmacists in guiding cannabis use. I don't think this section is adequately reflected in the title or the abstract and I think the text could be modified earlier on to state that a discussion of the pharmacist's role is included. This could also be reflected in the title as "pharmaceutical" in the title suggests just pharmacological considerations from my interpretation rather than patient-pharmacist interactions.

Line 441 - Monitoring blood levels for cannabis users is invasive and arguably not necessary when questionnaires on subjective effects will probably yield more useful information. That is probably worth mentioning.

Line 551 - Needs more capital letters for consistency "Category IV Future Implications and Recommendations for Cannabis Pharmaceutical Care in Mental Mealth"

Line 614 - "Yet, the lack of clinical evidence in published studies hinders..."

Comments on the Quality of English Language

The English is good. Very few errors which are mostly just typographical in nature. The few errors I noted I have included in the specific comments of my report.

Author Response

The changes have been made and the specific responses have been compiled into a document. Please see the attachment. Thank you.

Sincerely, The Authors.

Reviewer 2 Report

Comments and Suggestions for Authors

Thank you for the opportunity to review a topic, namely, ‘Exploring the Significance of Pharmaceutical Care in Mental Health: Spotlight on Cannabis’. This study aimed to evaluate pharmaceutical care in treating anxiety and depression alongside Cannabis use, focusing on safety and therapeutic efficacy optimization. 

Although the topic of manuscript is interesting, I have some concerns as follows: 

1. Lines 34-36: ‘Mental health issues such as anxiety and depression were identified as the most prevalent, comprising 31% and 28.9% of cases respectively, and are leading causes of disability worldwide’.

Suggestion: I suggest that the Authors should harmonize the terms, namely, 'anxiety symptoms' and 'depressive symptoms' throughout entire manuscript. For example, the above mentioned lines referred only to symptoms of these mental disorders.

2. Lines 34: ‘This narrative review aims to critically examine the role of pharmaceutical care in the...‘. I suggest that a 'narrative review' should be written in the both of abstract and title of the manuscript.

3. Line 80: The term ‘the guiding question’ could be changed to research question (RQ).

4. It is recommended for Authors to submit a PRISMA flowchart, too.

5. In the results section, it is doubtful whether the first table is appropriate as it contains data not included in the study. I suggest that the data included in the study should be presented in the results section.

6. Table 2 missing references. Please, add it.

7. Lines 252-253: ‘It has also been found that CBD has antidepressant effects at an effective dose of 30 mg.kg-1, provided there is activation of 5HT1A receptors‘.

Suggestion: The Authors should extend this information and describe experimental studies related to the dose and duration of administration of CBD.

8. What about side effects and safety? It is well known that chronic exposure to tetrahydrocannabinol (THC) can cause cannabis-related neurotoxic effects along with cognitive impairment, panic disorder, increased risk for developing depression, and other unwanted physical health problems.

9. The Authors draw too little reasoned conclusions. Whilst cannabis is used for therapeutic purposes and is legalized in the country concerned, the particular safe dose and duration of cannabis use as well as it's possible interactions with other medications are of paramount importance.

10. I suggest that the aim of this study should be specified and coupled with very specific research questions which must be answered clearly.

11. In this sensitive case, the non-experimental studies  such as cross-sectional, cohort, or case-control studies in design are not suitable for this narrative review. Therefore, in this case, this paper could include only the randomized controlled trials (RTCs)).

Author Response

The changes have been made and the specific responses have been compiled into a document. Please see the attachment. Thank you.

Sincerely, The Authors

Reviewer 3 Report

Comments and Suggestions for Authors

Introduction

Row 34: the paragraph refers to data from 2010 and 2019, but the reference (1) is from 2005.

Row 70-71: reference is required

Supplementary material: Title of the figure is missing.

Discussion

What were the search descriptors in the case of the four categories? E.g.: in the table 2, there are 13 articles mentioned.

Row 185-186: „ [17]. concluded that” „ [17] concluded that”

Row 253: “mg.kg-1” “mg x kg-1

There are studies about Cannabis anxiolytic and antidepressant effect, but nowadays standardized plant extracts are preferred compared to the plant materials, so it would be much more professionally appropriate discussing the efficacy of CBD and THC compounds in the treatment of anxiety and depression (mainly in terms of pharmaceutical care). The topic has a large literature (THC and anxiety and depression 221 results on PubMed and CBD and anxiety and depression 175 results).

Row 323: because of the fact that this review appears in 2024, this statement does not stand up, there are data about the metabolism and interaction of THC and CBD (e.g. DOI: 10.1208/s12248-021-00616-7)

Figure 1: instead of figure, tabulated format for presentation of these data is preferred (or improve the quality of the figure).

Row 462: instead of “well-known pharmacist” “prepared pharmacist” or “educated pharmacist” is suggested.

Conclusions:

It is true that healthcare professionals need reliable information about safety and efficacy of Cannabis, but the present review is too wide-ranging in terms of subject matter; a narrower area (e.g. Category 3 and 4) - discussed in more detail – would be more useful.

References: Article titles in English would help easier understanding

Materials and Methods: if no article meets the inclusion criteria, is recommended to discuss the topic from another perspective (as it is suggested earlier).

Author Response

(The authors gave the same response as above.)

Round 2

Reviewer 2 Report

Comments and Suggestions for Authors

The Authors answered the questions I raised.

Therefore, I recommend acceptation of the present paper.

Kind regards

Author Response

 Thank you very much for your response and we are very pleased with the opportunity to publish our article in this magazine

Reviewer 3 Report

Comments and Suggestions for Authors

The authors answered to all comments and reasonably defended their position on all comments.

A few more comments:

- Table 1 is preferred in "landscape" format to make the last column ("Exclusion Criteria") easier to red

- references 82-84 does not appear in the manuscript.

Author Response

Dear Reviewer and Editor,

All the requested revisions have been completed. I hope you consider this submission for acceptance in the journal.

Sincerely,